# Investigation of the Effects of Stress Hyperglycemia Ratio and Preoperative Computed Tomographic Angiography on the Occurrence of Acute Kidney Injury in Diabetic Patients following Surgical Thromboembolectomy

Orhan Guvenc [1], Mesut Engin [2,*], Filiz Ata [3] and Senol Yavuz [2]

[1] Departments of Cardiovascular Surgery, Medical Faculty of Uludağ University, 16310 Bursa, Turkey
[2] Department of Cardiovascular Surgery, University of Health Sciences, Bursa Yuksek Ihtisas Training and Research Hospital, 16310 Bursa, Turkey
[3] Department of Anesthesiology and Reanimation, University of Health Sciences, Bursa Yuksek Ihtisas Training and Research Hospital, 16310 Bursa, Turkey
* Correspondence: mesut.engin@saglik.gov.tr

**Abstract:** Acute lower extremity ischemia (ALI) is a cardiovascular emergency resulting from embolic and thrombotic causes. Although endovascular techniques have advanced, surgical thromboembolectomy is still the gold standard. Emergency thromboembolectomy surgery involves an ischemia-reperfusion injury, which also poses a risk for acute renal injury (AKI). The stress hyperglycemia rate (SHR) has recently emerged as an important prognostic value in emergency cardiovascular events. In the present study, we aimed to analyze the impact of preoperative contrast-enhanced tomographic angiography (CTA) and the SHR value on postoperative AKI in emergency thromboembolectomy procedures in patients with insulin-dependent diabetes mellitus (DM). In this retrospective analysis, patients with DM who received emergency surgical thromboembolectomy after being hospitalized at our hospital with ALI between 20 October 2015, and 10 September 2022, were included. Patients were classified into two groups: Group 1 ($N = 159$), who did not develop AKI, and Group 2 ($N = 45$), who did. The 45 patients in Group 2 and the 159 patients in Group 1 had median ages of 59 (39–90) and 66 (37–93), respectively ($p = 0.008$). The percentage of patients in Group 2 with Rutherford class IIB and admission times longer than 6 h was higher ($p = 0.003$, $p = 0.027$, respectively). To determine the variables affecting AKI after surgical embolectomy procedures, multivariate logistic regression analysis was used. In multivariate analysis Model 1, age > 65 years (odds ratio [OR]: 1.425, 95% confidence interval [CI]: 1.230–1.980, $p < 0.001$), preoperative high creatinine (OR: 4.194, 95% CI: 2.890–6.156, $p = 0.003$), and Rutherford class (OR: 0.874, 95% CI: 0.692–0.990, $p = 0.036$) were determined as independent predictors for AKI. In Model 2, age > 65 years (OR: 1.224 CI: 1.090–1.679, $p = 0.014$), preoperative high creatinine (OR: 3.975, 95% CI: 2.660–5.486, $p = 0.007$), and SHR (OR: 2.142, CI: 1.134–3.968, $p = 0.003$), were determined as independent predictors for amputation. In conclusion, when an emergency thromboembolectomy operation is planned in insulin-dependent DM patients, renal risky groups can be identified, and renal protective measures can be taken. In addition, to reduce the renal risk, according to the suitability of the clinical conditions of the patients, the decision to perform a CTA with contrast can be taken by looking at the SHR value.

**Keywords:** glucose; computed tomography; thrombosis; acute; contrast agent

## 1. Introduction

Acute lower extremity ischemia (ALI) is a cardiovascular emergency resulting from embolic and thrombotic causes. The diagnosis of acute extremity ischemia can be easily reached with a physical examination and a detailed anamnesis. According to the results of physical examination and Doppler ultrasonography, these patients can be operated on

without angiography before ischemia [1]. When these patients are diagnosed, surgical or endovascular procedures should be scheduled immediately. Although endovascular techniques have advanced, surgical thromboembolectomy is still the gold standard [2].

Emergency thromboembolectomy surgery involves an ischemia-reperfusion injury, which also poses a risk for acute renal injury (AKI). In addition, the administration of iodinated contrast material before surgery may also pose a risk for renal damage. In one study, it was shown that angiography increases the risk of renal damage in emergency trauma patients [3]. In another study, it was concluded that the time of angiography before coronary bypass did not affect postoperative renal failure in diabetic patients [4].

The stress hyperglycemia rate (SHR) has recently emerged as an important prognostic value in emergency cardiovascular events [5]. Studies have shown that the blood glucose (ABG) value alone after coronary interventions is predictive for the risk of renal damage after the procedure in patients without DM [6]. Therefore, it was thought that SHR value might be more predictive in DM patients. In a study conducted in this direction, it was shown that the SHR value calculated before coronary intervention in patients with acute myocardial infarction was a good predictor of postprocedural AKI [7].

In the present study, we aimed to analyze the impact of preoperative contrast-enhanced tomographic angiography (CTA) and the SHR value, which was calculated at the time of admission, on postoperative AKI in emergency thromboembolectomy procedures in patients with insulin-dependent diabetes mellitus (DM).

## 2. Materials and Methods

In this retrospective analysis, patients with DM who received emergency surgical thromboembolectomy after being hospitalized at our hospital with ALI between 20 October 2015 and 10 September 2022, were included. The hospital registry system and patient files were used to gather the patient data. All patients' demographic information, blood counts on admission, and postoperative problems were noted. Ultimately, after the exclusion criteria were met, 204 consecutive patients were enrolled in the research (Figure 1). After these procedures, patients were classified into two groups: Group 1 ($N = 159$), who did not develop AKI, and Group 2 ($N = 45$), who did.

### 2.1. Diagnosis and Treatment Strategy of ALI

A detailed history is taken, and a physical examination is performed on each patient who applies to our emergency department with a suspected ALI clinic. Doppler ultrasonography is also used to support the suspicion of ALI. As a result of these examinations, the diagnosis of ALI can be made, and surgical intervention can be planned [8]. However, there are also suggestions that surgery after CTA is beneficial by showing the thrombus location and arterial structure more objectively [9]. In our clinic, the rate of using CT imaging in patients with suspected ALI has increased approximately four times in the last four years.

After diagnosis, all patients were urgently hospitalized in our intensive care unit and taken into surgery. Local anesthetic and mild sedation were used during every procedure. After a linear incision in the femoral region, the common femoral, superficial femoral, and profunda femoral arteries were turned and suspended. After the arteriotomy to the common femoral artery, an embolectomy was performed with a Fogarty catheter (3–7 French). The procedure was continued until the thrombus material stopped coming forth, and the arteriotomy was primarily repaired when adequate distal and proximal flow was achieved. After the surgery, for at least one day, every patient was monitored in the intensive care unit under close monitoring. During this period, hourly heart rate examinations were performed. In medical treatments, after the heparin infusion (The activated clotting time stayed constant at 200–250 s) on the first day, a low molecular weight heparin (1 mg/kg, sc) treatment was applied for at least one week. In addition to these treatments, 100 mg/day of acetylsalicylic acid and clopidogrel 75 mg/day were given after the first day. In some patients, a re-embolectomy was determined according to the pulse findings in their close follow-up (a total of 12 patients).

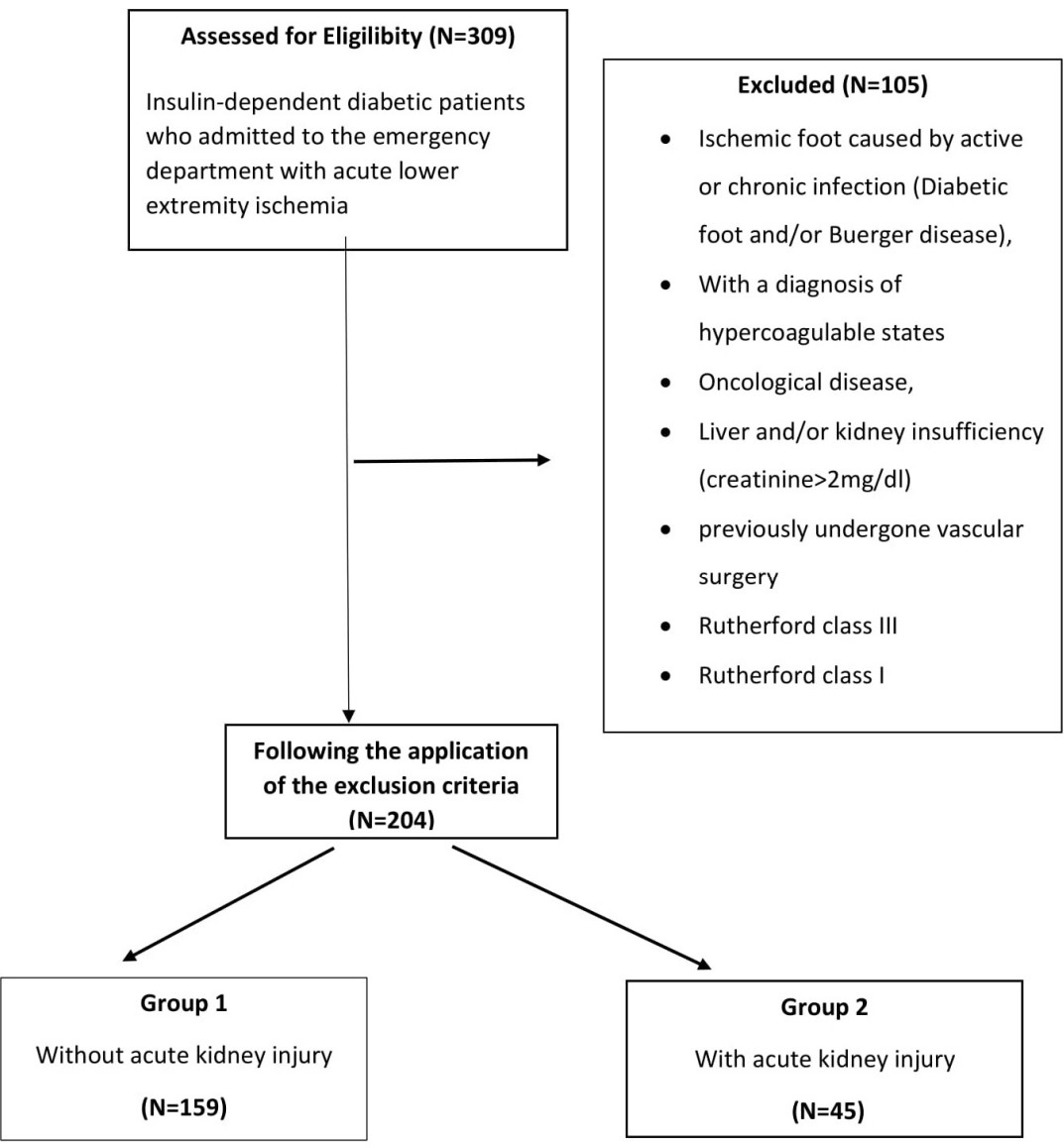

**Figure 1.** Flowchart of patient enrollment.

*2.2. Calculation of SHR*

Blood samples drawn from peripheral venous structures during hospitalization were used to determine the blood parameters for each patient. The next step was to acquire SHR using the following formula [5]:

$$\text{SHR} = \frac{\text{Admission blood glucose levels (mg/dL)}}{(28.7x \text{ glycosylared hemoglobin \%}) - 46.7} \quad (1)$$

*2.3. Diagnosis of AKI*

In the postoperative period, creatinine assessments were performed for three days in all patients. The postoperative renal injury will be performed according to the Kidney Disease Improving Global Outcomes (KDIGO) classification [10].

Three categories are used to group this information:

Stage 1: 1.5–1.9 times the baseline value or a rise of more than 0.3 mg/dL. Urine volume < 0.5 mL/kg/h for 6–12 h.

Stage 2: 2.0–2.9 times the initial value. Urine volume < 0.5 mL/kg/h for >12 h.

Stage 3: Three-fold rise from baseline, serum creatinine > 4.0 mg/dL, or the start of renal replacement treatment. Urine volume < 0.3 mL/kg/h for >24 h or anuria for >12 h.

*2.4. Statistical Analysis*

All data were evaluated using the SPSS program (IBM Corp., 2012, IBM SPSS Statistics for Windows, Version 21.0, Armonk, NY, USA: IBM Corp.). For categorical data, percentage and frequency values were determined; for continuous data, mean, standard deviation (SD), and median (minimum-maximum) values were computed. The Kolmogorov-Smirnov and Shapiro-Wilk tests were used to assess the distribution's normality. Data that were not normally distributed were evaluated using the Mann-Whitney U test and the Student's *t*-test, respectively. Nominal data were subjected to frequency and percentage analysis before the Chi-square test was used to compare them. To identify the predictors of post-operative renal injury, univariate logistic regression analysis was first performed. The multivariate logistic regression analysis included variables whose *p*-value fell below 0.25 in these assessments. Multivariate logistic regression analysis was carried out in two separate models; the model with the ABG value and the model with the SHR value. For the purpose of forecasting AKI, receiver-operating characteristic (ROC) curve analysis for ABG and SHR was carried out, and areas under the curve (AUC) were computed. The statistical significance of test results was defined as $p < 0.05$.

## 3. Results

Some 204 consecutive patients with insulin-dependent DM were included in the study. All patients underwent emergency surgical thromboembolectomy surgeries. The 45 patients in Group 2 and the 159 patients in Group 1 had median ages of 59 (39–90) and 66 (37–93), respectively ($p = 0.008$). The two groups shared similar characteristics with regard to the history of past cerebrovascular events, gender, hypertension, hyperlipidemia, rates of coronary artery disease and peripheral arterial disease, body mass index, and current smoking rates. The percentage of patients in Group 2 with Rutherford class IIB and admission times longer than 6 h was higher ($p = 0.003$, $p = 0.027$, respectively). Patients in the two groups also had similar vascular thrombus areas (Table 1).

**Table 1.** Demographic data of the patients.

| Variables | Group 1 (*N* = 159) | Group 2 (*N* = 45) | *p* Value |
|---|---|---|---|
| Age(years) | 59 (39–90) | 66 (37–93) | 0.008 ‡ |
| Female gender, *n* (%) | 44 (27.7%) | 14 (31.1%) | 0.652 * |
| Hypertension, *n* (%) | 86 (54.1%) | 27 (60%) | 0.481 * |
| Hyperlipidemia, *n* (%) | 51 (32.1%) | 19 (42.2%) | 0.206 * |
| CAD, *n* (%) | 54 (34%) | 17 (37.8%) | 0.635 * |
| PAD, *n* (%) | 30 (18.9%) | 8 (17.8%) | 0.868 * |
| Smoking, *n* (%) | 39 (24.5%) | 15 (33.3%) | 0.237 * |
| COPD, *n* (%) | 18 (11.3%) | 7 (15.6%) | 0.444 * |
| Previous CVE | 5 (3.1%) | 2 (4.4%) | 0.672 * |
| BMI (kg/m$^2$) | 24.9 (23.7- 35) | 25.1 (23.1–34) | 0.474 ‡ |
| FSDtoA time > 6 h | 19 (11.9%) | 11(24.4%) | 0.027 * |
| Occlusion region | | | 0.168 * |
| Iliofemoral occlusion, *n* (%) | 119 (74.8%) | 39 (64.4%) | – |
| Femoropoliteal occlusion, *n* (%) | 40 (25.2%) | 16 (35.6%) | – |
| Rutherford class, (%) | | | |
| IIA | 115 (72.3%) | 21 (48.9%) | 0.003 * |
| IIB | 44 (27.7%) | 23 (51.1%) | |
| Door-to operation, minutes | 75 (60–150) | 75 (60–180) | 0.674 ‡ |

* Chi-square test, ‡ Mann Whitney U test (Data is expressed as median (minimum-maximum)) CAD: Coronary artery disease, PAD: Peripheral arterial disease, BMI: Body mass index, CVE: Cerebrovascular event, COPD: Chronic obstructive pulmonary disease, FSDtoA: First symptom development to admission.

White blood cell, platelet, hematocrit, urea, albumin, HbA1c, and eAG levels did not significantly differ across the groups. ABG and SHR were considerably greater in Group 2 than in Group 1 (*p* = 0.017 and P0.001, respectively). Re-embolectomy (first 24 h) rates (3.1% versus 15.6%) and preoperative CTA usage rates (52.2% versus 71.1%) were also significantly higher in Group 2 (*p* = 0.002, *p* = 0.024, respectively) (Table 2).

**Table 2.** Preoperative laboratory and perioperative features of the patients.

| Variables | Group 1 (*N* = 159) | Group 2 (*N* = 45) | *p* Value |
|---|---|---|---|
| White blood Cell ($10^3/\mu$L) | 8.7 (7.9–15.9) | 9.1 (7.6–14.3) | 0.232 ‡ |
| Hematocrit (%) | 37 (29.1–51.8) | 39 (28–49.6) | 0.396 ‡ |
| Platelet ($10^3/\mu$L) | 228 (130–420) | 236 (120–390) | 0.275 ‡ |
| Creatinine, mg/dL | 1.1 (0.8–1.8) | 1.6 (0.7–1.9) | <0.001 ‡ |
| Urea, mg/dL | 20 (16–42) | 24 (14–36) | 0.147 ‡ |
| Albumin (g/L) | 36 (30.7–53.8) | 38 (31.6–51.9) | 0.489 ‡ |
| ABG, mg/dl | 270 (190–440) | 280 (210–436) | 0.017 ‡ |
| HbA1c, % | 7.7 (6.6–14.9) | 7.5 (6.8–13.6) | 0.237 ‡ |
| eAG, mg/dL | 144.8 (135–298) | 148.1 (132–286) | 0.181 ‡ |
| SHR | 1.59 (1.48–2.96) | 1.98 (1.71–3.12) | <0.001 ‡ |
| Re-embolectomy, *n* (%) | 5 (3.1%) | 7 (15.6%) | 0.002 * |
| Preoperative CTA, *n* (%) | 83 (52.2%) | 32 (71.1%) | 0.024 * |

* Chi-square test, ‡ Mann Whitney U test (Data is expressed as median (minimum-maximum)) ABG: Admission blood glucose, HbA1c: Hemoglobin A1c, eAG: estimated average glucose, SHR: Stress hyperglycemia ratio (ABG/eAG), CTA: Contrast-enhanced tomographic angiography.

The correlation between CTA use and AKI was examined using subgroup analysis. According to the research, only Stage 1 AKI (*p* = 0.015) is correlated with the usage of CTA (Table 3).

**Table 3.** Subgroup analysis for computed tomography usage and postoperative kidney injury stages.

| Variables | CTA (*N* = 115) | Non-CTA (*N* = 89) | *p* Value * |
|---|---|---|---|
| Stage 1 (N = 30) | 23 (20%) | 7 (7.8%) | 0.015 |
| Stage 2 (N= 12) | 8 (6.9%) | 4 (4.4%) | 0.459 |
| Stage 3 (N= 3) | 2 (1.7%) | 1 (1.1%) | 0.717 |

CTA: Computed tomographic angiography, * Chi-square test.

To determine the variables affecting AKI after surgical embolectomy procedures, multivariate logistic regression analysis was used (Table 4). In multivariate analysis Model 1, age > 65 years (odds ratio [OR]: 1.425, 95% confidence interval [CI]: 1.230–1.980, *p* < 0.001), preoperative high creatinine (OR: 4.194, 95% CI: 2.890–6.156, *p* = 0.003), and Rutherford class (OR: 0.874, 95% CI: 0.692–0.990, *p* = 0.036) were determined as independent predictors for AKI. In Model 2, age > 65 years (OR: 1.224 CI: 1.090–1.679, *p* = 0.014), preoperative high creatinine (OR: 3.975, 95% CI: 2.660–5.486, *p* = 0.007), and SHR (OR: 2.142, CI: 1.134–3.968, *p* = 0.003), were determined as independent predictors for amputation.

The effectiveness of ABG and SHR in predicting AKI following surgical thromboembolectomy procedures was assessed using ROC analysis. The cut-off value of ABG was 265 (AUC: 0.763, 95% CI: 0.680–0.846, *p* < 0.001, with 78.4% sensitivity and 56.7% specificity), and that of SHR was 1.89 (AUC: 0.849, 95% CI: 0.790–0.908, *p* < 0.001, with 86.4% sensitivity, 61.6% specificity) (Figure 2)

**Table 4.** Multivariate logistic regression analysis to show factors affecting the occurrence of postoperative acute kidney injury after surgical thromboembolectomy.

| Variables | Multivariate Analysis Model 1 | | | Multivariate Analysis Model 2 | | |
|---|---|---|---|---|---|---|
| | *p* | Exp(B) Odds Ratio | 95% CI Lower–Upper | *p* | Exp(B) Odds Ratio | 95% CI Lower Upper |
| Age > 65 years | <0.001 | 1.425 | 1.230–1.980 | 0.014 | 1.224 | 1.090–1.679 |
| Rutherford class IIb | 0.036 | 0.874 | 0.692–0.990 | 0.192 | 0.990 | 0.785–1.295 |
| Preoperative creatinine | 0.003 | 4.194 | 2.890–6.156 | 0.007 | 3.975 | 2.660–5.486 |
| Preoperative BTA | 0.341 | 1.332 | 0.875–1.692 | 0.278 | 1.190 | 0.819–1.532 |
| ABG, mg/dL | 0.211 | 1.443 | 0.838–2.157 | – | – | – |
| SHR | – | – | – | 0.003 | 2.142 | 1.134–3.968 |

ABG: Admission blood glucose, SHR: Admission blood glucose to estimated average glucose ratio.

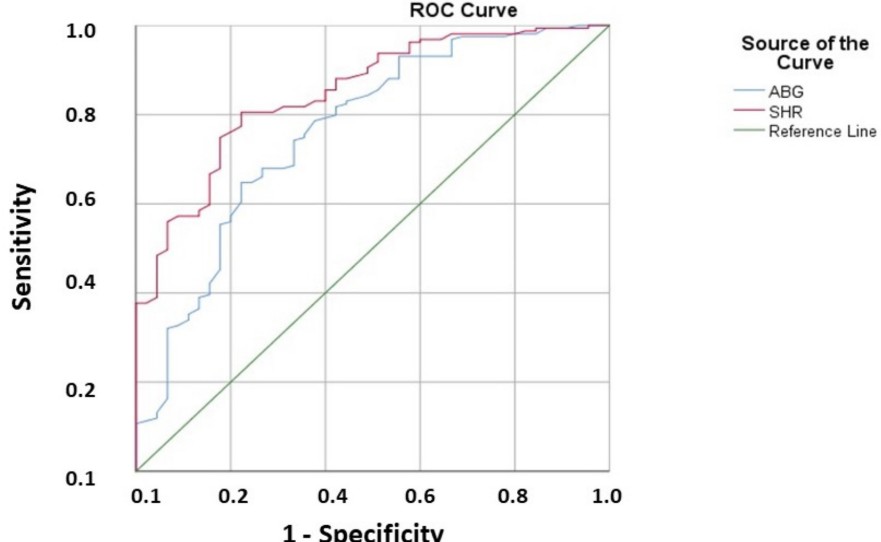

**Figure 2.** ROC (Receiver operation characteristic) curve and AUC (Area under the curve) for admission blood glucose (ABG) level and stress hyperglycemia ratio for predicting postoperative acute kidney injury. (ABG: Cut-off: 265, AUC: 0.763, 95% CI: 0.680–0.846, *p* < 0.001, 78.4% sensitivity and 56.7% specificity, SHR: Cut-off: 1.89, AUC: 0.849, 95% CI: 0.790–0.908, *p* < 0.001, 86.4% sensitivity and 61.6% specificity).

## 4. Discussion

ALI is an important clinical condition with mortal and morbid consequences, and surgery should be performed urgently on patients that are deemed to require it. Although these patients can be operated on after physical examination and DUSG findings, the use of CTA/conventional angiography has now become widespread. In the present study, we demonstrated that preoperative use of CTA in insulin-dependent DM patients increases the risk of postoperative AKI. However, we also showed that this increased risk was significant exclusively in Stage 1 AKI. In addition, we recently found that SHR, which is a valuable prognostic parameter in acute clinical conditions, is an independent predictor of the development of AKI after surgical thromboembolectomy.

On the impact of employing imaging techniques that require contrast agents before interventional surgeries, numerous research has been carried out. What causes conflict-induced AKI is not entirely understood. It has been suggested that several different systems must interact for AKI to occur. Among these mechanisms, it is commonly acknowledged that toxic effects on the tubular cells and a decrease in renal perfusion brought on by the direct action of contrast agents on the kidney are significant. The pathophysiological significance of the contrast agent's direct actions on tubular cells, as well as the other hypothesized

causes [11]. In a study by Yamamoto et al., including a large patient population, the effect of imaging with contrast agents on the development of AKI in emergency trauma patients was investigated. As a result of the analyses performed in this study, it was revealed that performing angiography in the emergency department doubled the risk of AKI [3]. In another study conducted by Doğan et al., patients with DM who underwent coronary artery bypass graft (CABG) operation were included. In this study, which included 421 patients retrospectively, the effect of CABG surgery timing after the coronary angiography on AKI was investigated. The patients were split into three groups. According to the moment, the CABG operation after coronary angiography (zero-three days, four-seven days, and >seven days) was performed. In this study, the authors did not find a significant relationship between angiography time and the development of AKI [4]. In our study, we investigated the effect of preoperative CTA on postoperative AKI in patients with DM who underwent emergency thromboembolectomy. By performing subgroup analysis, we also showed that Stage 1, 2, and 3 AKI development rates were high in patients who underwent preoperative CTA, but only the Stage 1 AKI development rate was statistically high.

Hyperglycemia occurs as an endocrine and metabolic response to acutely developing clinical conditions. This situation is detected at a rate of up to 30% in the hospitalized patient population, and this rate increases in elderly patients and patients with DM [12,13]. This condition, which occurs in response to acute events, increases oxidative stress and free radical production and leads to endothelial, vascular, and immune dysfunction [14]. In a study that included patients with DM with acute myocardial infarction (AMI), it was revealed that a high ABG value increased the risk of AKI [7]. In another study, it was revealed that hyperglycemia after coronary angiography increased the risk of AKI after the procedure [15]. In a study by Gorelik et al., the effect of ABG value on AKI in hospitalized patients was investigated. A significant relationship was shown between high ABG at admission and AKI and mortality in nondiabetic patients, and the authors concluded that this relationship decreased in DM patients [16]. In our study, the ABG value was higher in patients with AKI, nevertheless; multivariate analysis did not show that it was an independent predictor.

The ABG value is an important parameter as an indicator of the response to acute events in patients without DM. In a study conducted on intensive care patients, the SHR value was shown to be most useful in predicting poor outcomes in DM patients [17]. For this reason, studies investigating the predictive value of SHR value in DM patients have been conducted in the cardiovascular field. Accordingly, in a study conducted by Marenzi et al., the effect of acute and chronic glycemic status on AKI in AMI patients was investigated. This study included 474 consecutive diabetic AMI patients and showed that the SHR value was more reliable in predicting the risk of AKI than the ABG value alone [6]. In a study conducted by Ramon et al., 91 COVID-19 patients with DM who were hospitalized were included. The primary endpoints of the study were defined as the need for intensive care, the need for a mechanical ventilator, and mortality. Thirty-five (38.4%) patients had a primary outcome and in the analysis performed, a significant relationship was revealed between the primary outcome and the SHR value (hazard ratio: 1.57 95% CI 1.14–2.15, $p = 0.005$) [18]. In a study by Gao et al., the effect of SHR value on postprocedural AKI in AMI patients with DM was investigated. As a result of the multivariate analysis performed in this study, the SHR value was shown as an independent forecaster in predicting the risk of AKI (OR: 3.18, 95% CI: 1.99–5.09, $p < 0.001$) [7].

The Rutherford ALI classification is examined under four main headings according to the severity of ischemia (I, IIA, IIB, and III) [19]. In our study, we excluded patients who did not require emergency intervention (Rutherford I) and those with irreversible damage (Rutherford III). In the multivariate analysis Model 1 utilized in our study, we revealed that the severity of ischemia (Rutherford IIB versus IIA) increases the risk of postoperative AKI. As a result, the severity of ischemia may increase the risk of AKI, as it will increase oxidative stress due to ischemia-reperfusion [20]. In our study, being over the age of 65 was also shown as an independent predictor of AKI risk in both multivariate analysis models.

Fibrosis in cardiovascular structures increases with increasing age [21], and this fact may also increase the risk of AKI [22].

Although our study reached important results, there are also some limitations. First, we included a limited population of insulin-dependent DM patients. In addition, we conducted a single-center study which was planned retrospectively. However, we believe that our study will be a source for future multicenter prospective studies.

## 5. Conclusions

Acute limb ischemia is an important clinical condition that can lead to the loss of limbs as well as involve vital risks. AKI that occurs after these surgeries is also an important problem, as in many diseases. Our study showed that preoperative use of CTA can increase the risk of Stage 1 AKI in insulin-dependent DM patients. In addition, we showed the SHR value calculated from the blood values at the time of admission as an independent predictor of the development of AKI. Accordingly, when an emergency thromboembolectomy operation is planned in insulin-dependent DM patients, renal risky groups can be identified, and renal protective measures can be taken. In addition, to reduce the renal risk, according to the suitability of the clinical conditions of the patients, the decision to perform a CTA with contrast can be taken by looking at the SHR value. The findings of our study require to be supported by prospective randomized controlled studies.

**Author Contributions:** Conceptualization, O.G. and F.A.; methodology, O.G., M.E., S.Y. and F.A.; software, O.G.; validation, O.G. and M.E.; formal analysis, S.Y.; investigation, O.G. and M.E.; resources, O.G.; data curation, O.G.; writing—original draft preparation, O.G., M.E., S.Y. and F.A.; writing—review and editing, O.G., M.E., S.Y. and F.A.; visualization, O.G. and M.E.; supervision, O.G., M.E., S.Y. and F.A.; project administration, O.G.; funding acquisition, O.G. All authors have read and agreed to the published version of the manuscript.

**Funding:** This research received no external funding.

**Institutional Review Board Statement:** The study was approved by Bursa Yuksek Ihtisas Training and Research Hospital Clinical Research Ethics Committee (2011-KAEK-25 2022/09-05).

**Informed Consent Statement:** Written informed consent was obtained from the patients before their interventions.

**Data Availability Statement:** All data produced here are available and can be produced upon request.

**Conflicts of Interest:** The authors declare no conflict of interest.

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
