# Peer review of "Investigation of the Effects of Stress Hyperglycemia Ratio and Preoperative Computed Tomographic Angiography on the Occurrence of Acute Kidney Injury in Diabetic Patients following Surgical Thromboembolectomy"

_tomography, doi:10.3390/tomography9010020_

Round 1

Reviewer 1 Report

Comments are included in the manuscript as Notes in appropriate places.

Author Response

First of all, thanks for your attention and valuable suggestions.

Q1: Fig. 1 adds no additional information, it just repeats, although in a graphical manner, what was mentioned in the text.

A1: We referred to Figure 1 by shortening the clear spellings in the text.

Q2: correct expression is "low molecular weight heparin" (LMWH) and not ....height .... as written here.

A2: We edited this expression.

Q3: The name of a drug is missing !

A3: acetylsalicylic acid was added.

Q4: The degree of oliguria should be added to each stage, according to KDIGO classification (such as urine output less than 0.5 ml/kg/h for 6–12 hours for stage 1 etc.)

A4: The degree of oliguria was added to each stage.

Q5: The data are doubled - data presented in the text are presented again in table 4.

A5: To determine the variables affecting AKI after surgical embolectomy procedures, multivariate logistic regression analysis was used. We explained the results we obtained in the text and referred to Table 4.

Sincerely yours.

Reviewer 2 Report

Dear authors,

It is an interesting and timely article. I commend you for the idea of looking at AKI in these patients. And for writing the paper in a concise and clear style.

I have a question regarding line 105 - 106

"In addition to these treatments, 100 mg/day and clopidogrel 75 mg/day were given after the first day." – something is missing....100mg/day

Line 196: Perhaps you should comment  the statement about stage 1 AKI. How do you explain that the risk of developing stage I AKI is greater after exposure to iodinated substances? Possibly to evaluate the literature regarding this result.

Thank you and good luck!

Author Response

First of all, thanks for your attention and valuable suggestions.

Q1: I have a question regarding line 105 – 106 "In addition to these treatments, 100 mg/day and clopidogrel 75 mg/day were given after the first day." – something is missing....100mg/day

A1: acetylsalicylic acid was added. In addition to these treatments, 100mg/day of acetylsalicylic acid and clopidogrel 75mg/day were given after the first day

Q2: Line 196: Perhaps you should comment  the statement about stage 1 AKI. How do you explain that the risk of developing stage I AKI is greater after exposure to iodinated substances? Possibly to evaluate the literature regarding this result.

A2: The incidence of Stage 1, 2 and 3 renal failure was proportionally higher in patients using CTA. However, statistically, the incidence of Stage 1 renal failure was significantly higher in patients using CTA. To explain this more clearly, proportional values have been added to Table 3. In the next paragraph, we explained that the risk of developing AKI is greater after exposure to iodinated substances.

Sincerely yours.